# Neural Random Projection: From the Initial Task To the Input Similarity Problem

## Abstract

The data representation plays an important role in evaluating similarity between objects. In this paper, we propose a novel approach for implicit data representation to evaluate similarity of input data using a trained neural network. In contrast to the previous approach, which uses gradients for representation, we utilize only the outputs from the last hidden layer of a neural network and do not use a backward step. The proposed technique explicitly takes into account the initial task and significantly reduces the size of the vector representation, as well as the computation time. Generally, a neural network obtains representations related only to the problem being solved, which makes the last hidden layer representation useless for input similarity task. In this paper, we consider two reasons for the decline in the quality of representations: correlation between neurons and insufficient size of the last hidden layer. To reduce the correlation between neurons we use orthogonal weight initialization for each layer and modify the loss function to ensure orthogonality of the weights during training. Moreover, we show that activation functions can potentially increase correlation. To solve this problem, we apply modified Batch-Normalization with Dropout. Using orthogonal weight matrices allow us to consider such neural networks as an application of the Random Projection method and get a lower bound estimate for the size of the last hidden layer. We perform experiments on MNIST and physical examination datasets. In both experiments, initially, we split a set of labels into two disjoint subsets to train a neural network for binary classification problem, and then use this model to measure similarity between input data and define hidden classes. We also cluster the inputs to evaluate how well objects from the same hidden class are grouped together. Our experimental results show that the proposed approach achieves competitive results on the input similarity task while reducing both computation time and the size of the input representation.

## 1    Introduction

Evaluating object similarity is an important area in machine learning literature. It is used in various applications such as search query matching, image similarity search, recommender systems, clustering, classification. In practice, the quality of similarity evaluation methods depends on the data representation.

For a long time neural networks show successful results in many tasks and one such task is obtaining good representations. Many of these methods can be considered in terms of domain and task. The first case is when we have only unlabeled dataset. Then we can use autoencoders (Bank et al., 2020) or self-supervised approaches (Chen et al., 2020; Devlin et al., 2018; Doersch et al., 2015; Dosovitskiy et al., 2014; Gidaris et al., 2018; Noroozi & Favaro, 2016; Noroozi et al., 2017; Oord et al., 2018; Peters et al., 2018), which require formulation of a pretext task, which in most cases depends on the data type. These methods can be called explicit because they directly solve the problem of representation learning. Moreover, these models can be used for transfer knowledge when we have labeled data only in the target domain. The second case is where we have labeled data in the source and target domains. Then we can apply a multi-task learning approach (Ruder, 2017) or fine-tune the models (Simonyan & Zisserman, 2014; He et al., 2016) trained on a large dataset like ImageNet. Finally, there is the domain adaptation approach (Wang & Deng, 2018) where we have a single task but different source and target domains with labeled data in the target

domain (Hu et al., 2015) or with unlabeled data (Li et al., 2016; Yan et al., 2017; Zellinger et al., 2017).

In our study the target task is to measure similarity between objects and to define hidden classes based on it. We are interested in studying the issue of implicit learning of representations. Can the neural networks store information about subcategories if we don't explicitly train them to do this? More formally, we have the same source and target domains but different tasks and we don't have labeled data in the target domain. That makes our case different from the cases of transfer learning.

A solution to this problem could be useful in many practical cases. For example, we could train a model to classify whether messages are spam or not and then group spam campaigns or kind of attacks (phishing, spoofing, etc.) based on similarity measuring by trained neural network. Similar cases could be in the medicine (classifying patients into healthy/sick and grouping them by the disease) or in financial (credit scoring) area. The benefits are that we do not depend on the data type and, more importantly, we use only one model for different tasks without fine-tuning, which significantly reduces time for developing and supporting of several models.

Similar study was done in (Hanawa et al., 2020), where authors proposed evaluation criteria for instance-based explanation of decisions made by neural network and tested several metrics for measuring input similarity. In particular, they proposed the Identical subclass test which checks whether two objects considered similar are from the same subclass. According to the results of their experiments, the most qualitative approach is the approach presented in (Charpiat et al., 2019), which proposed to measure similarity between objects using the gradients of a neural network. In experiments, the authors applied their approach to the analysis of the self-denoising phenomenon. Despite the fact that this method has theoretical guaranties and does not require to modify the model to use it, in practice, especially in real-time tasks, using gradients tends to increase the computation time and size of vector representation. This approach will be described in more detail in Section 2. To avoid these problems, we propose a method that only uses outputs from the last hidden layer of a neural network and does not use a backward step to vectorize the input. In our research, we found that a correlation of neurons and insufficient width of the last hidden layer influence on the quality of representations obtained in implicit way. To solve these issues, we propose several modifications. First, we show that the weight matrix should be orthogonal. Second, we modify Batch-Normalization (Ioffe & Szegedy, 2015) to obtain the necessary mathematical properties, and use it with dropout(Srivastava et al., 2014) to reduce the correlation caused by nonlinear activation functions. Using orthogonal weight matrices allows us to consider the neural network in terms of Random Projection method and evaluate the lower bound of the width of the last hidden layer. Our approach will be discussed in detail in Section 3. Finally, in Section 4 we perform experiments on MNIST dataset and physical examination dataset (Maxwell et al., 2017). We used these datasets to show that our approach can be applied for any type of data and combined with different architectures of neural networks. In both experiments, we split a set of labels into two disjoint subsets to train a neural network for binary classification problem, and then use this model to measure the similarity between input data and define hidden classes. We also cluster the inputs to evaluate how well objects from the same class are grouped together. Our experimental results show that the proposed approach achieves competitive results on the input similarity task while reducing both computation time and the size of the input representation.

## 2 RELATED WORKS

Using a trained neural network to measure similarity of inputs is a new research topic. In (Charpiat et al., 2019) the authors introduce the notion of object similarity from the neural network perspective. The main idea is as follows: how much would parameter variation that changed the output for $x$ impact the output for $x'$? In principle, if the objects $x$ and $x'$ are similar, then changing parameters should affect the outputs in a similar way. The following is a formal description for one- and multi-dimensional cases of the output value of a neural network.

**One-dimensional case** Let $f_\theta(x) \in \mathbb{R}$ be a parametric function, in particular a neural network, $x$, $x'$ be input objects, $\theta \in \mathbb{R}^{n_\theta}$ - model parameters, $n_\theta$ - number of parameters. The authors proposed the following metric:

$$\rho_\theta(\boldsymbol{x}, \boldsymbol{x}') = \frac{\nabla_\theta f_\theta(\boldsymbol{x}') \nabla_\theta f_\theta(\boldsymbol{x})}{||\nabla_\theta f_\theta(\boldsymbol{x}')|| \, ||\nabla_\theta f_\theta(\boldsymbol{x})||} \qquad (1)$$

In this way, the object similarity is defined as the cosine similarity between the gradients computed at these points.

**Multi-dimensional case**    Let $f_\theta(x) \in \mathbb{R}^d$, $d > 1$. In this case, the authors obtained the following metric:

$$\rho_{\boldsymbol{\theta}, d}(\boldsymbol{x}, \boldsymbol{x}') = \frac{1}{d} Tr(K_{\boldsymbol{\theta}}(\boldsymbol{x}, \boldsymbol{x}')), \qquad (2)$$

where $K_\theta(\boldsymbol{x}, \boldsymbol{x}') = K_{\boldsymbol{x}', \boldsymbol{x}'}^{-1/2} K_{\boldsymbol{x}, \boldsymbol{x}'} K_{\boldsymbol{x}, \boldsymbol{x}}^{-1/2}$, and $K_{\boldsymbol{x}', \boldsymbol{x}} = \frac{\partial f_\theta}{\partial \theta}\Big|_{\boldsymbol{x}'} \frac{\partial f_\theta}{\partial \theta}^T\Big|_{\boldsymbol{x}}$ is calculated using the Jacobian matrix $\frac{\partial f_\theta}{\partial \theta}\Big|_{\boldsymbol{x}}$.

**Summary**    Unlike using third-party models for vectorization, such as VGG, this approach allows us to use any pre-trained neural network to calculate the similarity of input data. To achieve this, authors use gradients of neural network as illustrated in equation 1 and equation 2. This gradient-based solution takes into account all activations, which does not require the selection of a hidden layer for vectorization. However, it has a number of drawbacks. First of all, fast computation of gradients require additional computational resources. Second, the size of objects representation is $n_\theta$ for one-dimensional output and $n_\theta * d$ for multi-dimensional case. This means that increasing model complexity increases representation size and, as a result, can lead to large memory consumption. Motivation of this research is to develop an approach that reduces the size of the representation and uses only the forward pass of a trained neural network to work. In the next section, we will discuss the proposed approach in detail.

## 3    PROPOSED METHOD

In this paper, we suggest using outputs from the last hidden layer of a trained neural network to encode inputs. This representation has several advantages. First, the output of the last layer is usually low-dimensional, which allows us to get a smaller vector dimension. Second, and more importantly, on the last hidden layer the semantics of the original problem is taken into account to the greatest extent. For example, for a classification problem, data is theoretically linearly separable on the last hidden layer. Informally, this allows us to measure similarities within each class, which reduces false cases in terms of the original problem. This is extremely important for clustering spam campaigns, because this representation reduces the probability to group spam and legitimate messages together. However, as already mentioned in the section 1, a neural network that is trained to solve a simple problem does not retain the quality of representaions needed to solve a more complex one. Due to this fact, it's impossible to apply this representation as is. We have identified the main causes of this: a strong correlation of neurons and an insufficient size of the last hidden layer. Our goal is to avoid these in order to make the proposed vectorization reasonable. In practice, last hidden layers are usually fully connected. For this reason, we consider only this type of the last layer. In 3.1, we show how to reduce correlation between neurons, and in 3.2 we offer an estimate of the lower bound of the size of the last hidden layer and prove that the proposed representation can be used for the input similarity problem. We introduce the following notation:

1. $l \in \{0, 1, \ldots, L\}$ - layer number,
   where $l = 0$ - input layer (source space), $l = L$ - output layer, other - hidden layers.
2. $N_l$ - number of units in layer $l$
3. $\boldsymbol{h}_l \in \mathbb{R}^{N_l}$ - pre-activation vector
4. $\Phi(\boldsymbol{h}_l) = (\phi(\boldsymbol{h}_{l,1}), \ldots, \phi(\boldsymbol{h}_{l,N_l}))$ - activation vector
5. $\boldsymbol{W}_l \in \mathbb{R}^{N_{l-1} \times N_l}$, $\boldsymbol{b}_l \in \mathbb{R}^{N_l}$ - weight matrix and bias vector

### 3.1 NEURON CORRELATION

The benefits of decorrelated representations have been studied in (LeCun et al., 2012) from an optimization viewpoint and in (Cogswell et al., 2015) for reducing overfitting. We consider the decorrelated representations from information perspective. Basically, correlation of neurons means that neurons provide similar information. Therefore, we gain less information from observing two neurons at once. This phenomenon may occur due to the fact that the neural network does not retain more information than is necessary to solve a particular task. Thus, only important features are highlighted. The example in Fig. 1A illustrates that the output values of two neurons are linearly dependent, which entails that many objects in this space are indistinguishable. On the contrary (Fig. 1B), decorrelation of neurons provides more information and, as a result, the ability to distinguish most objects.

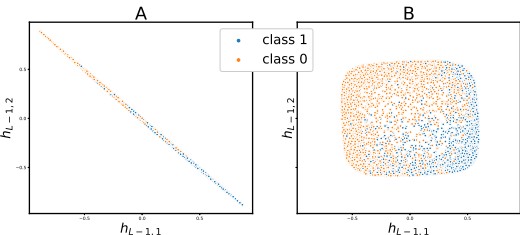

Figure 1: A) Correlated neurons. Only end task semantics are taken into account. B) Uncorrelated neurons. More objects are distinguishable

In the following paragraphs, we identify the main causes of the correlation of neurons and suggest ways to prevent it. We decided to consider the correlations before and after activation of neurons separately.

**Reducing correlation of neurons before activation**    Statement 1 explains the main reason for the correlation that occurs before activation.

**Statement 1.** *Suppose that for some layer $l$ the following conditions are satisfied:*

1. *$\mathbb{E}[\Phi(\boldsymbol{h}_l)] = 0$*

2. *$\mathbb{E}[\Phi(\boldsymbol{h}_l)^T \Phi(\boldsymbol{h}_l)] = \sigma_l^2 \boldsymbol{I}$ - covariance matrix as we required $\mathbb{E}[\Phi(\boldsymbol{h}_l)] = 0$*

*Then, in order for correlation not to occur on the layer $l + 1$, it is necessary that the weight matrix $\boldsymbol{W}_{l+1}$ be orthogonal, that is, satisfy the condition $\boldsymbol{W}_{l+1}^T \boldsymbol{W}_{l+1} = \boldsymbol{I}$*

The statement 1 shows that the first correlation factor on a fully connected layer is a non-orthogonal weight matrix, given that, if the input neurons do not correlate. See Appendix A.1 for proof of the statement 1. During training, the network does not try to maintain this property, solving the problem at hand. Later in this paper, we will add regularization to penalize the loss function if the weights are not orthogonal.

The corollary follows from the statement 1, which gives the second condition for preventing correlation. This corollary states that the dimension of the layer $l + 1$ should be no greater than on the layer $l$. Otherwise, this leads to a correlation and does not increase information. See Appendix A.1 for proof this corollary.

**Corollary 1.** *Suppose that the conditions of statement 1 are satisfied, then if the dimension $N_{l+1} > N_l$, then there is certainly a pair of neurons $\boldsymbol{h}_{l+1,i}, \boldsymbol{h}_{l+1,j}; i \neq j : \text{Cov}(\boldsymbol{h}_{l+1,i}, \boldsymbol{h}_{l+1,j}) \neq 0$*

It should be noted that there are also the studies addressing orthogonal weight matrices (Huang et al., 2017; Jia et al., 2019; Xie et al., 2017). However all of these works consider this topic from optimization perspective. In particular, in (Cho & Lee, 2017) was proposed an approach for optimization of loss on the Stiefel manifold $\mathcal{V}_p(\mathbb{R}^n) = \{\boldsymbol{W} \in \mathbb{R}^{n \times p} | \boldsymbol{W}^T \boldsymbol{W} = \boldsymbol{I}\}$ to ensure orthonormality of weight matrices throughout training. To achieve this, they applied the orthogonality regularization

(3) to require the Gram matrix of the weight matrix to be close to identity matrix. In our study we also use regularization (3) to ensure orthonormality of weight matrices.

$$\frac{1}{2}\sum_{l=1}^{L}\lambda_l \left|\left|\boldsymbol{W}_l^T\boldsymbol{W}_l - \boldsymbol{I}\right|\right|_F^2, \tag{3}$$

**Providing the necessary moments of neurons after activation**  In the statement 1, we relied on the zero expected value and the same variance of units in the layer. But the nonlinear activation function does not guarantee the preservation of these properties. Due to this fact, we cannot reason in the same way for the following layers. Therefore, we propose using an activation normalization approach similar to Batch-Normalization:

$$\hat{\phi}(\boldsymbol{h}_{l,i}) = \gamma_l \frac{\phi(\boldsymbol{h}_{l,i}) - \mu_{\phi(\boldsymbol{h}_{l,i})}}{\sqrt{\sigma^2_{\phi(\boldsymbol{h}_{l,i})} + \epsilon}}, \tag{4}$$

where $\gamma_l$ is a trainable scale parameter, $\mu_{\phi(\boldsymbol{h}_{l,i})}$, $\sigma^2_{\phi(\boldsymbol{h}_{l,i})}$ - parameters that are evaluated, as in Ioffe & Szegedy (2015). The difference compared to the standard Batch-Normalization is that the $\gamma_l$ is the same for all neurons and we removed the $\beta_{l,i}$ parameters. This leads to an expected value of zero and the same variance $\gamma_l^2$ of each unit in the layer.

**Reducing correlation of neurons after activation**  It should be noted that an activation function can also impact the formation of redundant features (Ayinde et al., 2019). In particular, in this work we use $tanh(x) \in (-1,1)$ as the activation function. There are several methods that prevent formation of redundant features. In (Cogswell et al., 2015) was proposed *DeCov loss* which penalizes non-diagonal elements of estimated covariance matrix of hidden representation. In (Desjardins et al., 2015; Blanchette & Laganière, 2018; Huang et al., 2018) were proposed approaches for learning decorrelation layers that perform the following transformation: $\tilde{\Phi}(\boldsymbol{h}_l) = (\Phi(\boldsymbol{h}_l) - \mu_{\Phi(\boldsymbol{h}_l)})\Sigma^{-\frac{1}{2}}_{\Phi(\boldsymbol{h}_l)}$. All of these methods have a common drawback: they require estimating covariance matrices. Often in practice the size of mini-batch is much smaller than is needed for estimating covariance matrices. Therefore, the covariance matrix is often singular. Moreover, methods that use the decorrelation layer are computationally expensive when it comes to high-dimensional embeddings, since it is necessary to calculate the square root of the inverse covariance matrix. This is especially evident in wide neural networks. Besides, these techniques add a significant amount of parameters $\sum_{l=1}^{L-1} N_l^2$.

As an alternative, we suggest using Dropout (Srivastava et al., 2014), which prevents units from co-adapting too much and reduces the correlation between neurons during training stage in the layer in proportion to $p$ - the probability of retaining a unit in the network. See Appendix A.2 for the proof of this statement.

It is important to note that we apply normalization to the input data (input layer $l = 0$) as well as Dropout, since it is not always possible to decorrelate data so that this does not affect the quality of training. Moreover, fully-connected layers are often used in more complex architectures, such as convolutional neural networks. Obviously, after convolution operations, transformed data will be correlated. In this case, we must apply dropout after the convolutional block in order to reduce the correlation of neurons and use the proposed approach for vectorization.

## 3.2 NEURAL NETWORK FROM THE RANDOM PROJECTION PERSPECTIVE

In the previous section, we proposed techniques of minimizing correlation between neurons. However, it is not guaranteed that the obtained representations are useful or sufficient for the similarity measuring of objects. In order to prove this, we consider a neural network as an application of the Random Projection method that allow us to estimate a lower bound of the representation size and define a metric to measure similarity.

The Random Projection method (Matoušek, 2013) is one of the methods of dimensionality reduction, which is based on the Johnson-Lindenstrauss lemma (Matoušek, 2013):

**Lemma 1.** *Let $\varepsilon \in (0,1)$ and $X = \{\boldsymbol{x}_1, \ldots, \boldsymbol{x}_n\}$ - a set of $n$ points in space $\mathbb{R}^d$; $k \geq \frac{C \log n}{\varepsilon^2}$, where $C > 0$ is a large enough constant. Then there exists a linear map $f : \mathbb{R}^d \to \mathbb{R}^k$ such that $\forall x, x' \in X$:*

$$(1 - \varepsilon)||\boldsymbol{x} - \boldsymbol{x}'|| \leq ||f(\boldsymbol{x}) - f(\boldsymbol{x}')|| \leq (1 + \varepsilon)||\boldsymbol{x} - \boldsymbol{x}'|| \tag{5}$$

As can be seen, Lemma 1 states only the existence of a linear map. However, there is also a probabilistic formulation of the Lemma 1, which states that if we take a sufficiently large $k$ and consider a random orthogonal projection onto the space $\mathbb{R}^k$, then inequality in equation 5 holds with high probability. This is the cornerstone of this work, allowing us to obtain Neural Random Projection, which is explained below.

In this work, we use the *tanh* activation function and assume that we are working in the linear region of the activation function. Due to this, we can make the following approximation:

$$\boldsymbol{h}_{L-1}(\boldsymbol{x}) \approx \boldsymbol{x}\tilde{\boldsymbol{W}}_1\tilde{\boldsymbol{W}}_2 \ldots \boldsymbol{W}_{L-1} + \tilde{\boldsymbol{b}} = x\hat{\gamma}\hat{\boldsymbol{W}} + \hat{\boldsymbol{b}} \tag{6}$$

where $\tilde{\boldsymbol{W}}_l$, $\tilde{\boldsymbol{b}}$ means that consistent use of scales and shifts was taken into account, respectively. Just $\boldsymbol{W}_{L-1}$ means that we consider the output before activation and hence before applying equation 4. And $\hat{\gamma}$ is common multiplier after all modified batch-normalizations. It is obvious that the final matrix $\hat{\boldsymbol{W}}$ is still orthonormal.

According to approximation in equation 6, pre-activation outputs of the last layer are an orthogonal projection of the input data. Moreover, the process of random orthogonal initialization and optimization on the Stiefel manifold can be seen as a random sampling of an orthogonal matrix. For this reasons, we consider the neural network from the point of view of the Random Projection method. Due to this fact, we use the similarity metric $L_2$ and get a lower bound estimate for the size of the last hidden layer ($k \geq \frac{C \log n}{\varepsilon^2}$), although in practice this estimate is often too conservative, since Lemma 1 does not take data structure into account. Therefore a small $C$ and $\varepsilon$ closer to one can be considered, as will be shown in the experiments section.

## 4 EXPERIMENTS

In this section we present the experiments on two datasets with different data types. Both experiments have the same structure. Initially, we group a set of labels into two disjoint subsets to train a neural network for binary classification problem, and then use this model to measure similarity between input data and define hidden classes. To evaluate the quality of defining hidden classes and hence the quality of similarity measure we use kNN classifier. To evaluate how well objects from the same hidden class are grouped together we use KMeans approach and *v-measure* (Rosenberg & Hirschberg, 2007). We use source data representation as baseline. To compare our approach, we consider 3 models (A, B, C) that have the same architectures but different regularizations in fully-connected layers. We also compare the representations from the last hidden layer with the previous gradient-based approach (Charpiat et al., 2019) using Model A, since the authors did not impose additional requirements on the neural network.

### 4.1 MNIST

We performed experiments on MNIST dataset, which consists of 70,000 images of size 28x28 with a 6-1 training-testing split. This dataset was chosen to show that our approach works with images and can be plugged-in after convolution blocks. See Appendix B.1 for more detailed description of the experiments.

**Results** As illustrated in Table 1, we achieved results comparable (kNN digit accuracy) with the previous approach and much better results from *v-measure* perspective. We also obtained a much smaller dimension of the representation (30 vs 14.9k), and in addition it is smaller than the original dimension (30 vs 784). This allowed us to drastically reduce the time for vectorization, using only the forward pass, and the time to measure the similarity of objects (kNN prediction time) and search

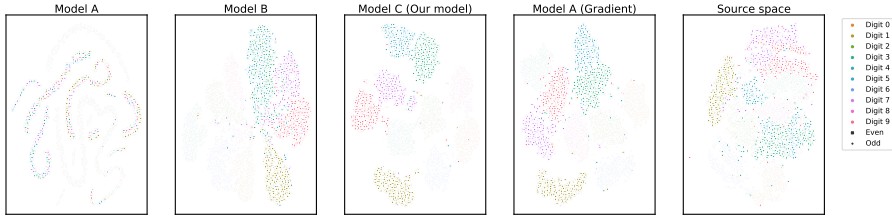

Figure 2: t-SNE last hidden layer (test set)

Table 1: Experimental results (MNIST dataset)

| Model (object rep.) | Model accuracy (even/odd) | kNN digit accuracy | V measure | Vec. time (s) | kNN prediction time (s) | Space dimension |
|---|---|---|---|---|---|---|
| A ($\nabla_\theta f_\theta$) | - | 96.8 | 0.542 | 677.04 | 188.79 | 14911 |
| A ($\boldsymbol{h}_{L-1}$) | 97.3±0.2 | 52.9±3.7 | 0.322±0.003 | 1.93±0.07 | 25.23±0.97 | 275 |
| B ($\boldsymbol{h}_{L-1}$) | 98.6±0.2 | 93.1±0.4 | 0.588±0.016 | 1.95±0.10 | 21.46±0.97 | 30 |
| C ($\boldsymbol{h}_{L-1}$) | **98.9±0.1** | **97.7±0.3** | **0.812±0.029** | 1.83±0.08 | 21.04±0.73 | 30 |
| $\boldsymbol{h}_0 = \boldsymbol{x}$ | - | 94.29 | 0.4 | - | 21.48 | 784 |

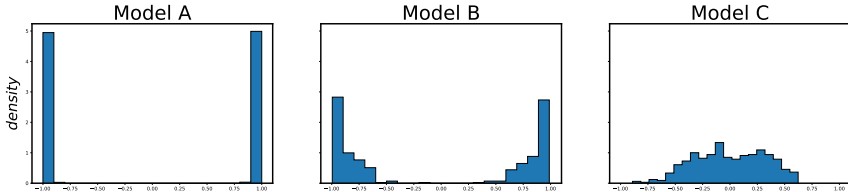

Figure 3: Distribution of correlations between pairs of neurons of the last hidden layer

Table 2: Spearman's correlattion between median of correlation distribution of neurons and target metrics

| $\varepsilon\,(\dim \boldsymbol{h}_{L-1})$ | KNN digit accuracy | V-measure |
|---|---|---|
| 0.2 (275) | -0.28 | -0.47 |
| 0.4 (68) | -0.86 | 0.13 |
| 0.6 (30) | -0.72 | -0.82 |
| 0.8 (17) | -0.57 | -0.81 |
| 1.0 (11) | -0.46 | -0.83 |

for the closest object. It should be noted that only our model (Model C) achieves comparable quality with the previous approach. Moreover we could improve evaluation of similarity in comparison to source representation.

Although the classic Batch-Normalization (Model B) gives a good increase in the quality of digit recognition compared to Model A, it still fails to achieve sufficient results on similarity task. This is well illustrated by the visualization of the representation of the last hidden layer (Fig. 2). As can be seen, Model A can only take into account the semantics of the original problem. In Model B, for the most part, the numbers are locally distinguishable, but some classes are strongly mixed, which is confirmed by the presence of high correlation (Fig. 3). Our model (Model C) can significantly reduce the correlation and gives an explicit separation with a large margin between the classes. In addition, the proposed modifications do not impair the quality of the model, as seen in Table 1 (Model accuracy).

As can be seen from Table 2, generally, there is a negative statistical relationship between target metrics and neuronal correlation.

## 4.2 HENAN RENMIN HOSPITAL DATA

Next, we used physical examination dataset (Maxwell et al., 2017), which contains 110,300 anonymous medical examination records. We retained only four most frequent classes (Normal, Fatty Liver, Hypertension and Hypertension & Fatty Liver), because this dataset is highly imbalanced. As a result, we got 100140 records. After that we split four classes on two corresponding groups "Sick" (Fatty Liver, Hypertension and Hypertension & Fatty Liver) and "Healthy" (Normal). After that we divided data set into train and test subsets in proportion 4-1.

It should be noted that this task is more difficult in comparison to the previous experiment because hidden classes are semantically related which makes it easier for the neural network to mix these classes into one.

This dataset was chosen to show that our approach works well with categorical features. Also we show that we can make the first layer wider than the source dimension and after that plug-in our approach. As mentioned above, the original classes are imbalanced so instead of accuracy metric we used $F_{1micro}$ score to evaluate the quality of the similarity measurement (kNN $F_{1micro}$). See Appendix B.2 for more detailed description of the experiments.

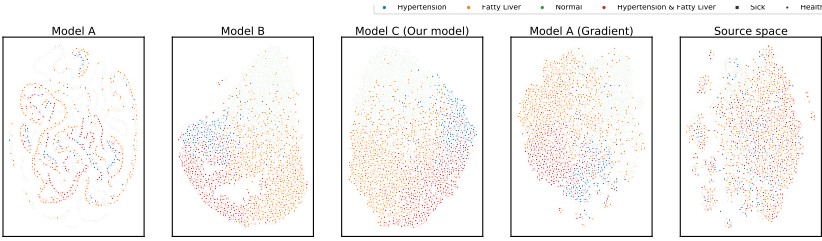

Figure 4: t-SNE last hidden layer (test set)

Table 3: Experimental results (Henan Renmin Hospital Data)

| Model (object rep.) | Model acc. (sick/ healthy) | kNN disease $F_{1micro}$ | V measure | Vec. time (s) | kNN pred. time (s) | Space dim |
|---|---|---|---|---|---|---|
| A ($\nabla_\theta f_\theta$) | - | 79.1 | 0.316 | 169.2 | 980.8 | 29111 |
| A ($h_{L-1}$) | **88.25±0.06** | 75.0±2.7 | 0.371±0.004 | 0.16±0.01 | 62.22±1.27 | 31 |
| B ($h_{L-1}$) | 86.7±0.1 | **79.6±0.2** | **0.411±0.015** | 0.28±0.02 | 93.26±2.59 | 31 |
| C ($h_{L-1}$) | 87.4±0.2 | **79.6±0.2** | 0.398±0.008 | 0.26±0.02 | 75.87±1.45 | 31 |
| $h_0 = x$ | - | 61.1 | 0.066 | - | 45.91 | 62 |

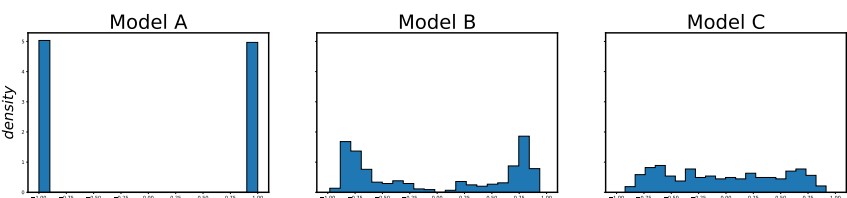

Figure 5: Distribution of correlations between pairs of neurons of the last hidden layer

**Results** As illustrated in Table 3, we achieved results comparable (kNN disease $F_{1micro}$ score) with the previous approach. As in the previous experiments we obtained a much smaller dimension

of the representation (31 vs 29k) and drastically reduced the time for vectorization and the time to measure the similarity of objects (kNN prediction time). It should be noted that we noticeably improved the quality of similarity measurement in comparison to the source space (61.1 vs 79.6) as can be seen also in Fig. 4. We also reduced correlation in comparison of the Model A and Model B, however it did not help us to obtain more quality representaions. This result can be explained by the fact that the hidden classes are strongly semantically related, in particular in "Sick" class. Therefore, it is easier for the neural network to mix these hidden classes into one. Probably, the data have more complicated structure in source space (see Table 3 for $h_0$) and it's not enough to only reduce the correlation. See Table 4 for detailed results.

Table 4: Spearman's correlattion between median of correlation distribution of neurons and target metrics

| $\varepsilon$ (dim $h_{L-1}$) | KNN $F_{1micro}$ disease | V-measure |
|---|---|---|
| 0.2 (282) | -0.83 | -0.6 |
| 0.4 (70) | -0.81 | -0.13 |
| 0.6 (31) | -0.11 | -0.34 |
| 0.8 (17) | 0.1 | -0.1 |
| 1.0 (11) | 0.02 | -0.15 |

As can be seen from Tabel 4, in this case, the statistical relationship between neuron correlation and target metrics is not so strong. Especially for the small dimension of the last hidden layer. This may be due to the fact that if the layer size is insufficient, there are not enough dimensions to describe the variability of the data, even if the data is not highly correlated.

## 5 CONCLUSION

In this work, we studied an approach for obtaining implicit data representation. In order to obtain implicit data representation, we introduced the Neural Random Projection method, which includes regularization of the neural network in the form of optimization among orthogonal matrices, a modification of Batch-Normalization and its combination with Dropout. This allowed to obtain representations applicable for input similarity measure using trained neural network. We experimentally compared our approach to the previous one and showed that it significantly reduced both computation time and the size of the input representation. Finally, our approach allowed us to introduce the $L_2$ metric on the representations and improve the quality of the similarity measurement in comparison with the source space. And how can be seen from the experiments the correlation and the size of the last hidden layer not yet all factors affecting on the final implicit representation. This remains an open question, and in the future we are planning to research it more. We expect that our study will be useful for many applied problems, where it is initially very difficult to determine a method for measuring data similarity in its original form.

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

# A   THEORETICAL PART

## A.1   REDUCING CORRELATION OF NEURONS BEFORE ACTIVATION

**Statement 1.** *Suppose that for some layer l the following conditions are satisfied:*

*1.* $\mathbb{E}[\Phi(\boldsymbol{h}_l)] = 0$

*2.* $\mathbb{E}[\Phi(\boldsymbol{h}_l)^T \Phi(\boldsymbol{h}_l)] = \sigma_l^2 \boldsymbol{I}$ *- covariance matrix as we required* $\mathbb{E}[\Phi(\boldsymbol{h}_l)] = 0$

*Then, in order for correlation not to occur on the layer $l + 1$, it is necessary that the weight matrix $\boldsymbol{W}_{l+1}$ be orthogonal, that is, satisfy the condition $\boldsymbol{W}_{l+1}^T \boldsymbol{W}_{l+1} = \boldsymbol{I}$.*

*Proof.* To prove this, we consider the expected value and the covariance matrix of the vector $\boldsymbol{h}_{l+1}$, using the relation $\boldsymbol{h}_{l+1} = \Phi(\boldsymbol{h}_l)\boldsymbol{W}_{l+1} + \boldsymbol{b}_{l+1}$ and let the condition of orthogonality be satisfied:

1. Expected value:
$$\mathbb{E}[\boldsymbol{h}_{l+1}] = \mathbb{E}[\Phi(\boldsymbol{h}_l)\boldsymbol{W}_{l+1} + \boldsymbol{b}_{l+1}] = \mathbb{E}[\Phi(\boldsymbol{h}_l)]\boldsymbol{W}_{l+1} + \mathbb{E}[\boldsymbol{b}_{l+1}] = \boldsymbol{b}_{l+1}$$

2. Covariance matrix:
$$\mathbb{E}[(\boldsymbol{h}_{l+1} - \mathbb{E}[\boldsymbol{h}_{l+1}])^T(\boldsymbol{h}_{l+1} - \mathbb{E}[\boldsymbol{h}_{l+1}])] = \mathbb{E}[\boldsymbol{W}_{l+1}^T \Phi(\boldsymbol{h}_l)^T \Phi(\boldsymbol{h}_l)\boldsymbol{W}_{l+1}] =$$
$$= \boldsymbol{W}_{l+1}^T \mathbb{E}[\Phi(\boldsymbol{h}_l)^T \Phi(\boldsymbol{h}_l)]\boldsymbol{W}_{l+1} =$$
$$= \sigma_l^2 \boldsymbol{W}_{l+1}^T \boldsymbol{W}_{l+1} = \sigma_l^2 \boldsymbol{I}$$

$\square$

**Corollary 1.** *Suppose that the conditions of statement 1 are satisfied, then if the dimension $N_{l+1} > N_l$, then there is certainly a pair of neurons $\boldsymbol{h}_{l+1,i}, \boldsymbol{h}_{l+1,j}$; $i \neq j$ : $\text{Cov}(\boldsymbol{h}_{l+1,i}, \boldsymbol{h}_{l+1,j}) \neq 0$*

*Proof.* Suppose, towards a contradiction, that $N_{l+1} > N_l$ and $\forall i, j \in \{1, \dots, N_{l+1}\}$, $i \neq j \Rightarrow \text{Cov}(\boldsymbol{h}_{l+1,i}, \boldsymbol{h}_{l+1,j}) = 0$. By the statement 1, this is possible only if $\boldsymbol{W}_{l+1}$ is an orthogonal matrix, which means that there is a linearly independent system of $N_{l+1}$ vectors of dimension $N_{l-1}$. It follows that rank $\boldsymbol{W}_{l+1} \geq min(N_l, N_{l+1}) = N_l$, which is impossible, therefore the matrix $\boldsymbol{W}_{l+1}$ is not orthogonal, thus we have a contradiction. $\square$

## A.2   REDUCING CORRELATION OF NEURONS AFTER ACTIVATION

For further discussion, we describe the Dropout method in our notation: $\boldsymbol{z}_l = \boldsymbol{r}_l \cdot \hat{\Phi}(\boldsymbol{h}_l)$, where $\boldsymbol{r}_l = (\boldsymbol{r}_{l,1}, \dots, \boldsymbol{r}_{l,N_l})$; $\boldsymbol{r}_{l,i} \sim Bernoulli(p)$ and the $\cdot$ sign means element-wise multiplication, $p$ the probability of retaining a unit in the network and $\hat{\Phi}$ denotes activation after modified Batch-Normalization. The following statement explains how Dropout reduces correlation.

**Statement 2.** *Dropout reduces the correlation in proportion to $p$*

*Proof.* Let $C_{ij}$, $i \neq j$ - correlation value between neurons $i, j$. Given the properties of $\hat{\phi}(\boldsymbol{h}_{li})$, we obtain the following expression for the correlation: $C_{ij} = \frac{\mathbb{E}[\hat{\phi}(\boldsymbol{h}_{li}), \hat{\phi}(\boldsymbol{h}_{lj})]}{\gamma^2}$. Now consider the correlation after applying Dropout:

$$\hat{C}_{ij} = \frac{\mathbb{E}[\boldsymbol{z}_{li}, \boldsymbol{z}_{lj}]}{\sqrt{\mathbb{E}[\boldsymbol{z}_{li}^2]\mathbb{E}[\boldsymbol{z}_{lj}^2]}} = \frac{\mathbb{E}[\boldsymbol{r}_{li}\hat{\phi}(\boldsymbol{h}_{li}), \boldsymbol{r}_{li}\hat{\phi}(\boldsymbol{h}_{lj})]}{\sqrt{\mathbb{E}[\boldsymbol{r}_{li}^2\hat{\phi}^2(\boldsymbol{h}_{li})]\mathbb{E}[\boldsymbol{r}_{lj}^2\hat{\phi}^2(\boldsymbol{h}_{lj})]}} =$$

$$= \frac{\mathbb{E}[\boldsymbol{r}_{li}]\,\mathbb{E}[\boldsymbol{r}_{lj}]\,\mathbb{E}[\hat{\phi}(\boldsymbol{h}_{li}), \hat{\phi}(\boldsymbol{h}_{lj})]}{\sqrt{\mathbb{E}[\boldsymbol{r}_{li}^2]\,\mathbb{E}[\boldsymbol{r}_{lj}^2]\,\mathbb{E}[\hat{\phi}^2(\boldsymbol{h}_{li})]\mathbb{E}[\hat{\phi}^2(\boldsymbol{h}_{lj})]}} =$$

$$= \frac{p^2 \mathbb{E}[\hat{\phi}(\boldsymbol{h}_{li}), \hat{\phi}(\boldsymbol{h}_{lj})]}{p\gamma^2} = pC_{ij}$$

$\square$

Here we used the fact that $\boldsymbol{r}_{li}, \boldsymbol{r}_{lj}$ are independent Bernoulli random variables, and they are also independent of $\hat{\phi}(\boldsymbol{h}_{li}), \hat{\phi}(\boldsymbol{h}_{lj})$.

# B   DETAILS OF EXPERIMENTS

**Computing infrastructure**   All experiments were performed on Intel(R) Xeon(R) CPU E5-2650 0 @ 2.00GHz, 189GB RAM and CentOS Linux 7 (Core) x86-64 operating system. The models were implemented using Python 3.6v and TensorFlow 2.2.0v library.

## B.1   MNIST

**Experiments description**

1. Models (A, B, C) trained to solve binary classification problem (even/odd digit). Number of epochs - 100, optimizer - *Adam* with learning rate 0.001; batch size - 256; validation split - 0.1, early stopping: min delta - 1e-3 and patience 5 epochs.

2. kNN classifier trained ($k = 9$) to solve digit classification problem, using pre-trained **model A** and metric from equation 1 from previous approach. We measured vectorization time on the complete dataset and evaluated digit accuracy, even/odd accuracy and prediction time of this kNN model on test set.

3. Everything is similar to the previous item, only the representations of the last hidden layer of each model and the $L_2$ metric were used.

4. Like previous item, only the source space representation and the $L_2$ metric were used.

We used models with the following base architecture:

$Conv2D(16 \times 3 \times 3) \rightarrow ReLU \rightarrow MaxPolling2D(2 \times 2) \rightarrow Conv2D(8 \times 3 \times 3) \rightarrow ReLU \rightarrow MaxPolling2D(2 \times 2) \rightarrow Dense(64) \rightarrow tanh \rightarrow Dense(\dim h_{L-1}) \rightarrow tanh \rightarrow Dense(1)$

The size of the last hidden layer was chosen corresponding to the lower bound from section 3.2:

$$\dim h_{L-1} \approx \frac{\log 6 \cdot 10^4}{\varepsilon^2}$$

In all models, only $l_2$ with $\lambda = 0.01$ regularization was used in the convolution block. Further, the difference between models in fully-connected layers will be given.

**Model A**: simple network, only $l_2$ with $\lambda = 0.01$ regularization is used. And the variable ranges from 0.2 to 1.0 with step 0.2.

**Model B**: orthogonal matrix initialization, regularization from equation 3 with $\lambda_l \in \{0.0, 0.01, 0.1, 1\}$ and $l_2$ regularization on bias vector; standard Batch-Normalization is used before activation and Dropout with $p \in \{0.0, 0.1, 0.2, 0.3\}$ after activation.

**Model C** (proposed model): modified Batch-Normalization is used after activation and Dropout after Batch-Normalization.

Each model with each combination of hyperparameters is trained 20 times. It is needed to obtain $95\%$ confident intervals for target metrics (KNN accuracy digit and v-measure).

## B.2   HENAN RENMIN HOSPITAL DATA

**Experiments description**

1. Models (A, B, C) trained to solve binary classification problem (sick/healthy). Number of epochs - 200, optimizer - *Adam* with initial learning rate 0.001 and exponential decay factor 0.99 every 500 iterations; batch size - 128; validation split - 0.1, early stopping: min delta - 1e-3 and patience 5 epochs.

2. kNN classifier ($k = 9$) trained to classify type of disease, using pre-trained **model A** and metric from equation 1. We measured vectorization time on the complete dataset. And we evaluated $F_{1micro}$ score on hidden classes because they are imbalanced, sick/healthy accuracy and prediction time of this kNN model on test set.

3. Everything is similar to the previous item, only the representations of the last hidden layer of each model and the $L_2$ metric were used.

4. Like previous item, only the source space representation and the $L_2$ metric were used.

We used models with the following base architecture: $Dense(128) \rightarrow tanh \rightarrow Dense(96) \rightarrow tanh \rightarrow Dense(64) \rightarrow tanh \rightarrow Dense(32) \rightarrow tanh \rightarrow Dense(\dim \boldsymbol{h}_{L-1}) \rightarrow Dense(1)$

The size of the last hidden layer was chosen corresponding to the lower bound from previous section:

$$\dim h_{L-1} \approx \frac{\log 8 \cdot 10^4}{\varepsilon^2}$$

Each model with each combination of hyperparameters is trained 20 times. It is needed to obtain $95\%$ confident intervals for target metrics (KNN $F_{1micro}$ disease and v-measure).

**Model A**: simple network, only $l_2$ with $\lambda = 0.001$ regularization is used.

**Model B**: orthogonal matrix initialization, regularization from equation 3 with $\lambda_l \in \{0.0, 0.01, 0.1, 1\}$ except the first layer and $l_2$ regularization on bias vector; standard Batch-Normalization is used before activation and Dropout with $p \in \{0.0, 0.1, 0.2, 0.3\}$ after activation.

**Model C** (proposed model): modified Batch-Normalization is used after activation and Dropout with after Batch-Normalization.

