# OpenReview forum: "Neural Random Projection: From the Initial Task To the Input Similarity Problem"
_ICLR.cc/2021/Conference — Reject_

### Official Review · AnonReviewer4 · 2020-10-27
**Neural decorrelation for dimensionality reduction?**

**Rating:** 7
**Confidence:** 3

**Review:**

I love the problem of neural information decorrelation which is so widespread in neural systems precisely for the reasons that the authors point out to detect novel information.
Despite the paper is nicely written, it’s perhaps not best organized, because I had to go back and forth to find out the main author’s contributions. I think the main contribution of the authors is combining random projection initialization, training with orthonormality regularization (as proposed by others) and drop out. The authors also claim that they can reduce the dimensionality by applying Lemma 1.

The results in Fig. 3 illustrate how the model achieves decent to batch normalization but with a flatter distribution of correlations, which I agree with the authors is the goal.

The paper could benefit from a small summary at the introduction that lists with clarity the components of the model in a succinct manner.

The three models compared should be explained better in the experimental section. So, people can read the paper faster.

I’d be interesting to know what component on the model is more important and for what. Is it the regularization, is it the random projection initialization used for faster convergence and information compression? Is it the dropout necessary to ensure a flatter distribution of correlations? An experimental section illustrating the impact on the model performance with each of these components would help us learn more.

---

> ### Author Response · Authors · 2020-11-24
> **Response to the reviewer**
>
> Thank you very much for take a time to review our article.
>
> We conducted much more experiments, tried to research how hyperparameters  influence on correlations.  Unfortunately we haven't more time to research this interesting question.
> We suggest that orthogonal regularization influence more than dropout referring to statement 1 from our work.

---

### Official Review · AnonReviewer1 · 2020-10-29
**on the benefits of orthogonal weight matrices**

**Rating:** 4
**Confidence:** 4

**Review:**

The paper studies the usage of the representations developed in the last layer of a neural network as a way to measure the similarity between input patterns. The fundamental idea revolves around the concept of orthogonal weight matrices, to decorrelate the activations of the neurons, and which would definitely enrich the internal representations developed by the neurons in the last hidden layer. In the paper, it is suggested to regularize during training to maintain the orthogonality of the weight matrices. Moreover, a variant of Batch Normalization is proposed. The method proposed in the paper is then experimentally applied and evaluated on two benchmark datasets (MNIST and  Henan Renmin).

PROs:
- The study of intrinsic richness in neural network architectures related to the shape of the weight matrices, in this case orthogonality, is of great appeal.

CONs:
- In the experimental part I could not see (even in the appendix and in the code) any consideration made in relation to the hyper-parameters of the neural networks, and of the used learning algorithms. The choices made in the paper (and reported in the appendix), e.g. on the learning rate, on the number of epochs, on the regularizer coefficients, seems rather arbitrary and would be better selected on a validation set.
- Being (at least in my view) the paper based on suggesting and using orthogonality as a design choice for the weight matrices in a neural network, the novelty content seems rather limited
- Personally, I found the manuscript a bit hard to follow in some parts, limiting the overall readability. For example, the notation is first introduced in the appendix (and I suggest to introduce it at least before section 3.1). Moreover, I suggest to re-phrase some overstating sentences, e.g. on page 3 where it is reported that " [...] there is usually a significant information loss on the last hidden layer, which makes it impossible to apply this representation as is. We have identified the main causes of this: a strong correlation of neurons and an insufficient size of the last hidden layer. [...]". While the intuition is clear here, I could not see in the paper a clear evidence of the fact that the information loss in the last hidden layer is caused by the correlation of small-sized layers.

-- EDIT: I am thankful to the authors for their insightful answer to my concerns, and for the though work in reviewing the manuscript. At this stage, I would keep my score (also in light of the other comments), but I hope that the authors find the suggestions from all the reviewers useful to re-present a more consolidated work soon.

---

> ### Author Response · Authors · 2020-11-24
> **Response**
>
> >>In the experimental part I could not see (even in the appendix and in the code) any consideration made in relation to the hyper-parameters of the neural networks, and of the used learning algorithms. The choices made in the paper (and reported in the appendix), e.g. on the learning rate, on the number of epochs, on the regularizer coefficients, seems rather arbitrary and would be better selected on a validation set.
>
>
>
> Thank you for your comment. We tried to improve our work and expanded our experiments, in particular we made a grid of hyperparameters and set early stopping for epochs. Our experiments continue to show that our approach performs better than other models.
> >>Being (at least in my view) the paper based on suggesting and using orthogonality as a design choice for the weight matrices in a neural network, the novelty content seems rather limited
>
>
>
> The use of orthogonal matrices is not a suggestion for solving the problem, but a consequence of allowing to reduce the correlation of output neurons for fully connected layers.
>
>
>
> >>I suggest to re-phrase some overstating sentences, e.g. on page 3 where it is reported that " [...] there is usually a significant information loss on the last hidden layer, which makes it impossible to apply this representation as is. We have identified the main causes of this: a strong correlation of neurons and an insufficient size of the last hidden layer. [...]". While the intuition is clear here, I could not see in the paper a clear evidence of the fact that the information loss in the last hidden layer is caused by the correlation of small-sized layers.
>
>
>
> Thanks. We have rephrased this passage and added an experiment that shows that the size of the last hidden layer and the correlation of neurons affect the quality of the representation.

---

### Official Review · AnonReviewer5 · 2020-11-06
**too many handwavy claims...**

**Rating:** 3
**Confidence:** 4

**Review:**

The paper proposed two methods for reducing the correlation of neurons within a layer so that the effective dimension of the last layer can be reduced as well, and the two methods include enforcing orthogonality on weight matrices, and batchnorm after activation functions, There are many issues regarding the claims and the experiments presented in the paper.

1. Is the paper trying to propose a novel way of measuring the similarity between inputs or to improve the vector representations generated from the last layer to be more reflective and indicative in terms of the fine-grained structure of the data? Those two topics seemed intertwined with each other, and also it seemed that the authors are using one as the supporting argument for the other.

For the first one, the authors claimed that the reasons for the information loss from bottom to the top layer in a neural networks include (1) correlation of neurons, and (2) insufficient width of the last hidden layer.

Let's assume that both reasons are valid, then why bother using neural networks for comparing similarity between datasets? One can certainly consider kernel methods for comparison, and specifically kernel two-sample tests with characteristic or universal kernels.

IMO, I don't agree with the reasons listed by the authors. In fact, let's take VGGNet and ResNet34 pretrained on ImageNet for comparison. The dimension of the top layer of VGGNet is 4096, while it is 512 for ResNet34, and as known, ImageNet classification task has 1000 different labels. It means that those 1000 512-D vectors learnt in the top layer in ResNet34 are correlated with each other to some degree, while those 1000 4096-D vectors from VGGNet could be orthogonal to each other. However, as shown, ResNet34 generalises better than VGGNet not only on ImageNet tasks, but also on other vision-related tasks. In this comparison, clearly the architecture matters more than the dimension of the last layer and the correlation of neurons.

For the second one, let's consider a linear system, ridge regression in particular. By adding a certain level of l2 regularisation on the parameter vector or matrix, it effectively moves the covariance matrix of the input data close to an identity matrix, which improves the orthogonality of the covariance matrix. It seems that adding l2 regularisation is a simpler way than ones proposed in this paper to reduce the correlation of neurons.

2. The example presented in Fig 1. is not a valid piece of evidence for the claims made in this paper.

From the perspective of information theory, the definition of information is the thing/observation that reduces the uncertainty. It heavily depends on the subject of the stufy. Therefore, from classification perspective, one can say that, in Fig. 1a, the model has observed enough amount of information so that the uncertainty of the two neurons' behaviours is drastically reduced, while Fig. 2b still presents a large amount of uncertainty. The paper referred to the term 'information' many times without consolidating specific occasions.

3. The activation function applied in the experiments is tanh, which squashes values to be in between -1 and 1, and the authors proposed to apply BatchNorm after tanh activation function. I don't see how the batchnorm after tanh is helpful in anyways as both mean and variance of the neurons are bounded already.

4. The comparison conducted on the MNIST dataset seemed missing some fundimental vectorisation and quantisation methods, including both learning-based or non-learning-based. If the goal was to find a low-D vector representation of data, one can consider many approaches without using neural networks. This comment goes back to my first point in a way that I am not super clear about what the paper is trying to solve.

---

> ### Author Response · Authors · 2020-11-16
> **Some questions to the reviewer**
>
> Thank you very much for the reviewing our paper. We will be glad to discuss all of unclear moments.
>
> 1. "Is the paper trying...?"
> In this work, we are interested in studying the issue of implicit learning of representations for obtaining hidden classes. Can the neural networks store information about subcategories if we don’t explicitly train them to do this?
>
> A solution to this problem could allow us to use only one model for different tasks without fine-tuning, which significantly reduces time for developing and supporting of several models.
>
> 2. You said that why bother using neural networks for comparing similarity between datasets?
>
>     However, in our work, we don't compare similarity between datasets. We have the same source and target domains but different tasks and we don’t have labeled data in the target domain.
>
>     Could you explain please, what did you mean?
>
> 3. About the example with pre-trained VGGNet and ResNet34 on ImageNet dataset.
>
>     You said, "... 1000 512-D vectors of ResNet34 are correlated with each other" and "... 1000 4096-D vectors of VGGNet are orthogonal".
>
>     Could you please clarify which vectors you are considering?
>
>     And in this regard, we don't understand how the vectors of weight matrix might be correlated and how you compare such properties as correlation and orthogonal together. Furthermore, in this work, we operate with the output of the last hidden layer.
>
>     Could you explain please, what did you mean? Unfortunately, at the moment we don't understand how your argument is related with our work.
>
> 4. About $l_2$ -regularization.
>
>     First of all, considering the solution of ridge regression $w = (X^TX +  aI)^{-1}X^Ty$ we see that the adding of $l_2$-regularization doesn't moves covariance matrix to identity matrix. It is needed to obtain not singular matrix. And, consequentely, this allow us to compute the inverse matrix. The second, in model A only $l_2$-regularization is used and how can be seen from Fig.3 and Fig.5 the features are highly correlated. Your statement contradicts our experimental data.
>
> 5. "The example presented in Fig 1. is not a valid..."
>
>     You said that from classification perspective, one can say that, in Fig. 1a, the model has observed enough amount of information so that the uncertainty of the two neurons' behaviours is drastically reduced, while Fig. 2b still presents a large amount of uncertainty.
>
>     However, in our work we say a similar thing ("... This phenomenon may occur due to the fact that the neural network does not retain more information than is necessary to solve a particular task", page 3). And example in Fig. 1 shows that, a model whose neurons are correlated (Fig 1a) only retain the information for classification. The model (Fig. 1b) can not only solve the same problem, but also distinguish between input objects within the same class. This follows from the fact that we have retained more uncertainty, and therefore we get more information from observation.
>
>     We agree with your statement, but it is not entirely clear to us why it contradicts our statements.
>
> 6. "The tanh activation function is already bounded..."
>
>     In our work we don't rely on bounded mean and variance. We require the certain properties that are mentioned on the page 5 in the paragraph "Providing the necessary moments of neurons after activation".
>
>     Could you please clarify your point of view?
>
> 7. About the experiments on the MNIST dataset.
>
>     How it was mentioned above, our goal is to obtain representations by implicit approach.
>     We believe that comparisons should be made within methods that are not explicitly trained to represent data.
>
> At this point, several of your arguments remain unclear to us. We would be glad to continue discussing our work.

---

### Decision · Program_Chairs · 2021-01-07
**Final Decision**

**Decision:**

Reject

**Comment:**

Techniques are introduced for improving representation learning
capabilities of neural networks, and the result is interpreted in
terms of random projections.

In further discussion, even the reviewer with the highest grade said
that the paper does not yet have enough clarity to address the
reviewers' comments. Particularly important would be to isolate the
causal impact of the proposed components in the final result. But also
several technical details would need to be clarified including
comparing to simple l2 regularization and precise implications of Fig
1.

Positive aspects: The problem of learning representations and
decorrelation is of course important. The authors have imagination,
and the authors are encouraged to improve the ideas by taking the
reviewer feedback into account.